# Impact of scaling up dolutegravir on antiretroviral resistance in South Africa: A modeling study

Anthony Hauser[1], Katharina Kusejko[2], Leigh F. Johnson[3], Huldrych F. Günthard[2,4], Julien Riou[1], Gilles Wandeler[1,5], Matthias Egger[1,3,6]*, Roger D. Kouyos[2,4]*

1 Institute of Social and Preventive Medicine, University of Bern, Bern, Switzerland, 2 Division of Infectious Diseases and Hospital Epidemiology, University Hospital Zurich, University of Zurich, Zurich, Switzerland, 3 Centre for Infectious Disease Epidemiology and Research, University of Cape Town, South Africa, 4 Institute of Medical Virology, University of Zurich, Zurich, Switzerland, 5 Department of Infectious Diseases, Bern University Hospital, University of Bern, Bern, Switzerland, 6 Population Health Sciences, Bristol Medical School, University of Bristol, Bristol, United Kingdom

* matthias.egger@ispm.unibe.ch (ME); roger.kouyos@uzh.ch (RDK)

**Data Availability Statement:** All code and manuscript are available from https://github.com/anthonyhauser/MARISA2. In addition, the values of all the model parameters are reported in S1 Text.

## Abstract

### Background

Rising resistance of HIV-1 to non-nucleoside reverse transcriptase inhibitors (NNRTIs) threatens the success of the global scale-up of antiretroviral therapy (ART). The switch to WHO-recommended dolutegravir (DTG)-based regimens could reduce this threat due to DTG's high genetic barrier to resistance. We used mathematical modeling to predict the impact of the scale-up of DTG-based ART on NNRTI pretreatment drug resistance (PDR) in South Africa, 2020 to 2040.

### Methods and findings

We adapted the Modeling Antiretroviral drug Resistance In South Africa (MARISA) model, an epidemiological model of the transmission of NNRTI resistance in South Africa. We modeled the introduction of DTG in 2020 under 2 scenarios: DTG as first-line regimen for ART initiators, or DTG for all patients, including patients on suppressive NNRTI-based ART. Given the safety concerns related to DTG during pregnancy, we assessed the impact of prescribing DTG to all men and in addition to (1) women beyond reproductive age; (2) women beyond reproductive age or using contraception; and (3) all women. The model projections show that, compared to the continuation of NNRTI-based ART, introducing DTG would lead to a reduction in NNRTI PDR in all scenarios if ART initiators are started on a DTG-based regimen, and those on NNRTI-based regimens are rapidly switched to DTG. NNRTI PDR would continue to increase if DTG-based ART was restricted to men. When given to all men and women, DTG-based ART could reduce the level of NNRTI PDR from 52.4% (without DTG) to 10.4% (with universal DTG) in 2040. If only men and women beyond reproductive age or on contraception are started on or switched to DTG-based ART, NNRTI PDR would

The patient data from the HIV cohort studies are confidential and cannot be shared publicly.

**Funding:** This study was supported by the National Institutes of Health (www.nih.gov) (National Institute of Allergy and Infectious Diseases and the Eunice Kennedy Shriver National Institute of Child Health and Human Development; grant number 2U01AI069924 to ME) and the Swiss National Science Foundation (Grant No. 174281 to ME). RDK was supported by the Swiss National Science Foundation (Grant number BSSGI0_155851). The funders had no role in study design, data collection and analysis, decision to publish, or preparation of the manuscript.

**Competing interests:** I have read the journal's policy and the authors of this manuscript have the following competing interests: HFG has received unrestricted research grants from Gilead Sciences and Roche; fees for data and safety monitoring board membership from Merck; consulting/ advisory board membership fees from ViiV, Gilead Sciences, Sandoz and Mepha. ME is a member of the Editorial Board of PLOS Medicine. The other authors have declared that no competing interests exist.

**Abbreviations:** 3TC/FTC, lamivudine/emtricitabine; ART, antiretroviral therapy; DTG, dolutegravir; MARISA, Modeling Antiretroviral drug Resistance In South Africa; MSM, men who have sex with men; NAMSAL, New Antiretroviral and Monitoring Strategies in HIV-infected Adults in Low-income countries; NNRTI, non-nucleoside reverse transcriptase inhibitor; NRTI, nucleoside reverse-transcriptase inhibitor; NTD, neural tube defects; OR, odds ratio; PI, protease inhibitor; PDR, pretreatment drug resistance; TDF, tenofovir; TLD, Tenofovir, Lamivudine, and Dolutegravir.

reach 25.9% in 2040. Limitations include substantial uncertainty due to the long-term predictions and the current scarcity of knowledge about DTG efficacy in South Africa.

## Conclusions

Our model shows the potential benefit of scaling up DTG-based regimens for halting the rise of NNRTI resistance. Starting or switching all men and women to DTG would lead to a sustained decline in resistance levels, whereas using DTG-based ART in all men, or in men and women beyond childbearing age, would only slow down the increase in levels of NNRTI PDR.

## Author summary

### Why was this study done?

- The scale-up of antiretroviral therapy (ART) in resource-limited settings has achieved an unprecedented reduction in HIV-related morbidity and mortality.

- The success of ART is however threatened by increasing levels of resistance to antiretroviral drugs of the non-nucleoside reverse transcriptase inhibitors (NNRTI) class.

- Replacing NNRTIs by dolutegravir (DTG) may curb the spread of resistance, but it is unclear how effective this switch will be and which patient groups should be switched from NNRTI to DTG.

- It has been debated whether DTG should be given to women, because of a potential risk of birth defects, and to patients already on an NNRTI-based therapy.

### What did the researchers do and find?

- Using a mathematical model simulating the HIV epidemic in South Africa, we find that scaling up DTG-based ART can halt the increase of NNRTI resistance.

- This predicted effect of DTG depends crucially on including both women and people already on NNRTI-based ART among patients to whom DTG will be prescribed.

- Restricting DTG to men or to patients initiating ART would substantially reduce its potential to curb resistance at the population level, as in this case it could merely slow down but not halt the spread of NNRTI resistance.

- Patients still relying on NNRTI-based therapy would in this case face increased risk of resistance and therapy failure.

### What do these findings mean?

- Our model highlights the potential of DTG scale up to curb NNRTI resistance.

- In order to halt the increase in NNRTI resistance, DTG should become accessible to both women and people currently on NNRTI-based therapy.

## Introduction

The rollout of antiretroviral therapy (ART) in South Africa is estimated to have prevented 0.73 million HIV infections between 2004 and 2013 as well as 1.72 million deaths between 2000 and 2014 [1,2]. However, the spread of non-nucleoside reverse transcriptase inhibitor (NNRTI)-resistant viruses is threatening this success [3]. An estimated 16% of AIDS-related deaths and 8% of ART costs will be attributable to HIV drug resistance up to 2030 in the sub-Saharan African countries that reached HIV pretreatment drug resistance (PDR) levels above 10% in 2016 [4].

In Southern Africa, dolutegravir (DTG), an integrase inhibitor drug, is being introduced on a large scale as part of fixed-dose combinations of Tenofovir, Lamivudine, and Dolutegravir (TLD) [5]. With a high genetic barrier to resistance, DTG has the potential to curb the spread of antiretroviral resistance, as it is highly effective, well tolerated, and affordable in resource-limited settings [6–9]. Mathematical models explored the effectiveness and cost-effectiveness of prescribing DTG to all ART initiators [10]. These models found that the introduction of DTG was cost-saving and reduced HIV mortality in people living with HIV who initiate ART [10].

The introduction of DTG has been complicated by the increased risk of neural tube defects (NTD) in women living with HIV using DTG at the time of conception [11] and other potential side effects such as weight gain [9,12]. Concerns surrounding NTD risk have delayed the rollout of DTG and, in some settings, led to recommending DTG-based regimens only for men and women who are not at risk of pregnancy [13,14]. For South Africa, a mathematical modeling study showed that DTG-based first-line ART for all women of child-bearing potential would prevent more deaths among women and more sexual HIV transmissions than either NNRTI-based ART for women of child-bearing potential or women without contraception, but increase pediatric deaths [15]. In its 2019 guidelines, WHO recommends DTG in combination with nucleoside reverse-transcriptase inhibitors (NRTIs) for first-line ART, with the proviso that "women should be provided with information about benefits and risks to make an informed choice regarding the use of DTG" [16].

It is likely that in many settings, people living with HIV on NNRTI-based first-line ART will be switched to DTG-based ART; however, the rate of the transition will vary between countries and settings. For second-line ART, WHO recommends DTG-based ART in people living with HIV for whom an NNRTI-based first-line regimen has failed [16]. Again, the rate of switching to DTG-based second-line ART will vary, influenced by concerns about the development of DTG resistance in patients who switch with preexisting resistance to NRTIs [17]. Taken together, it is likely that for the foreseeable future, a considerable fraction of people living with HIV, and particularly women, may continue to rely on NNRTI-based ART regimens, even in the case when guidelines recommend DTG. In this context, NNRTI resistance will likely remain an important issue during and after the rollout of DTG.

We adapted the MARISA model (Modeling Antiretroviral drug Resistance In South Africa) [18] to predict the impact of different scenarios regarding the scale-up of DTG-based ART on NNRTI PDR (NNRTI resistance in the remaining of this article) in South Africa for 2020 to 2040.

## Materials and methods

### Extended MARISA model

Described in detail elsewhere [18], MARISA is a deterministic compartmental model of both the general HIV epidemic and the NNRTI resistance epidemic in South Africa. It consists of 4 dimensions representing (1) care stages (see Fig 1); (2) disease progression according to the CD4 cell counts; (3) sex; and (4) NNRTI resistance. Care stages distinguish between infected, diagnosed, and treated individuals (either with NNRTI- or protease inhibitor (PI)-based regimen), with subsequent treatment-specific suppression (*Supp* compartment in Fig 1) or failure (*Fail* compartment). *Treat init.* compartments represent individuals treated for less than 3 months.

We simulated the adapted model from 2005 to 2040 assuming the rollout of DTG-based ART started in 2020 under different scenarios (see below). We further assumed that all men and a proportion $p_1$ of women are eligible for DTG, this proportion varied between scenarios. Before 2020, an NNRTI is used in first-line ART and PIs in second-line regimens. As first-line regimen, DTG is prescribed from 2020 on either to ART initiators (i.e., eligible and ART-naive people living with HIV) or for switching to DTG-based first-line ART (i.e., people on NNRTI and eligible for DTG). We assumed that patients failing DTG are switched to a PI-based regimen. For DTG-ineligible women, the cascade of care remains unchanged after 2020 (Fig 1).

### Calibration and extension of the MARISA model

We previously calibrated the MARISA model by combining different sources of data. Rates related to either treatment response (NNRTI- or PI-based regimen) or disease progression (characterized by CD4 counts) were estimated using clinical data from data from 5 cohorts in South Africa (Aurum Institute, Hlabisa, Kheth'Impilo, Rahima Moosa, and Tygerberg) that participate in the IeDEA collaboration [19]. These cohorts provided longitudinal information

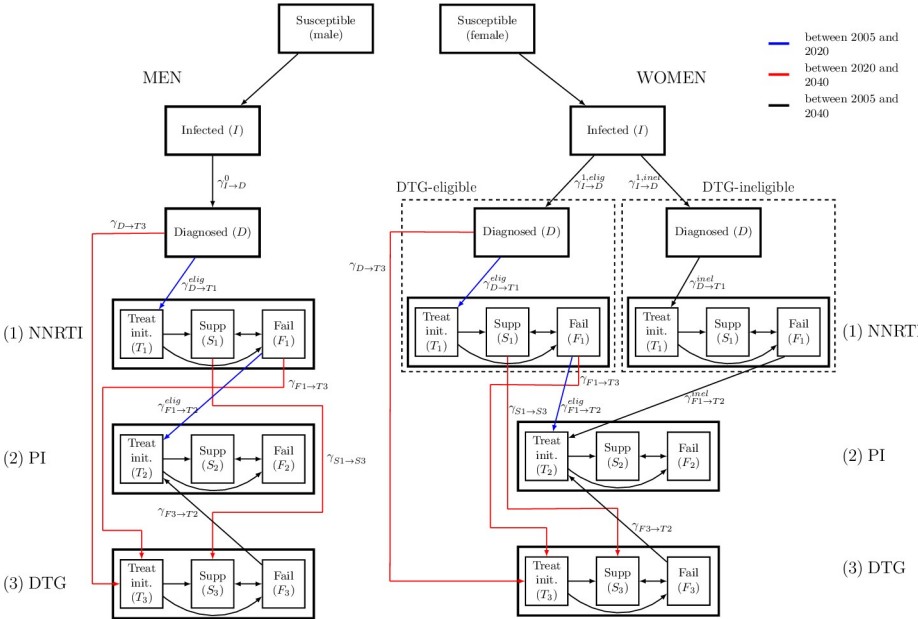

**Fig 1. The adapted MARISA model.** The model differentiates DTG-eligible from DTG-ineligible women. The model structure related to the cascade of care is shown. DTG, dolutegravir; MARISA, Modeling Antiretroviral drug Resistance In South Africa; NNRTI, non-nucleoside reverse transcriptase inhibitor; PI, protease inhibitor.

for 30,317 HIV-infected adults. Other parameters were either estimated from the literature (e.g., resistance-related parameters) or fitted to estimates from the Thembisa model (e.g., diagnosis rates and treatment initiation rates). Thembisa is a demographic projection model on which the official UNAIDS estimates for South Africa are based [20]. More details about the calibration procedure can be found in [18] and in the Supporting information (S1 Text, Section 1.1).

We added and modified parameters in order to model the introduction of DTG. We assumed that the DTG initiation rate $\gamma_{D \to T3}(t)$ is the same as the NNRTI initiation rate $\gamma_{D \to T1}(t)$ from 2020. Both NNRTI and DTG initiation rates increase until 2022, as a consequence of the Treat-All policy that was implemented in 2017. From 2022 onwards, they are assumed to remain constant (S1 Text, Section 2.2). Finally, we fixed switching rates from NNRTI- to DTG-based regimens for both eligible suppressed (see "Scenarios") and all failing individuals ($\gamma_{S_1 \to S_3}(t)$ and $\gamma_{F_1 \to T_3}(t)$, respectively) to 1 year$^{-1}$. We assumed that suppressed individuals would stay suppressed when switching, while failing individuals would start DTG in the *Treat init.* compartment (see Fig 1). The main parameters are summarized in Table 1.

In the adapted MARISA model, we also added a fifth NRTI resistance dimension to model the impact of NRTI resistance on DTG-efficacy. NRTI resistance is defined as having resistance to both tenofovir (TDF) and lamivudine/emtricitabine (3TC/FTC), the 2 backbones that are usually combined with DTG (see S1 Text, Section 2.5) [21]. We assume that NRTI resistance is acquired when failing a NNRTI-based regimen. In view of the low levels of NRTI PDR that are observed in Africa and its rapid reversion to wild type, we assumed as an approximation that it cannot be transmitted. We investigated different impacts of NRTI resistance on DTG-based regimen and different DTG-efficacies. For this aim, we recalibrated the model so that it reflects different odds ratios (ORs) of DTG-failure between NRTI-resistant and NRTI-susceptible individuals (OR = 1, OR = 2, OR = 5). In the main analysis, we assumed that DTG-

**Table 1. Main parameters used in the extended MARISA model.**

| Rate | Description | Source/Definition |
|---|---|---|
| *Rates estimated in the previous MARISA model [18]* | | |
| $\gamma_{I \to D}^{k,elig/inel}$, $\gamma_{D \to T_1}^{k,elig}$ | Diagnosis rate, treatment initiation rate to NNRTI | Calibrated by fitting MARISA to Thembisa model (see [18]) from 2005 to 2016 (see S1 Text Section 2.2) |
| $\gamma_{T_1 \to S_1}, \gamma_{T_1 \to F_1}, \gamma_{F_1 \to S_1}, \gamma_{S_1 \to F_1}$ | NNRTI suppression and failure rates | Estimated with individual epidemiological data from IeDEA-SA [19] (see S1 Text Table B) |
| $\gamma_{T_2 \to S_2}, \gamma_{T_2 \to F_2}, \gamma_{F_2 \to S_2}, \gamma_{S_2 \to F_2}$ | PI suppression and failure rates | Estimated with data from IeDEA-SA [19] (see S1 Text Table B) |
| $\gamma_{F_1 \to T_2}^{k,elig/inel}$ | Rate of switching from NNRTI to PI (before 2020) | Estimated with data from IeDEA-SA and then adjusted (see S1 Text Section 2.2) |
| *Added rates* | | |
| $\gamma_{T_3 \to S_3}, \gamma_{T_3 \to F_3}, \gamma_{F_3 \to S_3}, \gamma_{S_3 \to F_3}$ | DTG suppression and failure rates | Calibrated with data from NAMSAL study [12] (see S1 Text Section 2.5) |
| $\gamma_{D \to T3}(t)$ ($t \geq 2020$) | DTG initiation rate (from 2020) | Same DTG initiation rate as for NNRTI (for DTG-inel. people) $\gamma_{D \to T3} := \gamma_{D \to T1}^{inel}$ |
| $\gamma_{I \to D}^{k,elig}(t), \gamma_{I \to D}^{k,inel}(t), (t \geq 2020)$ | Diagnosis rate from 2020 (distribution across DTG-eligibility classes) | $\gamma_{I \to D}^{1,elig}(t) = p_1 \cdot \gamma_{I \to D}^1(2020)$ and $\gamma_{I \to D}^{1,inel}(t) = (1 - p_1) \cdot \gamma_{I \to D}^1(2020)$, for $t \geq 2020$ (see S1 Text Section 2.2) |
| $\gamma_{F_1 \to T_2}^{elig/inel}(t), (t \geq 2020)$ | Rate of switching from NNRTI to PI (DTG-inel.) | $\gamma_{F_1 \to T_2}^{inel}(t) = \gamma_{F_1 \to T_2}(t)$, $\gamma_{F_1 \to T_2}^{elig}(t) = 0$, for $t \geq 2020$ (see S1 Text Section 2.2) |
| $\gamma_{F_3 \to T_2}(t), (t \geq 2020)$ | Rate of switching from DTG to PI | $\gamma_{F_3 \to T_2}(t) := \gamma_{F_1 \to T_2}^{inel}(t), t \geq 2020$ |
| $\gamma_{S_1 \to S_3}(t), (t \geq 2020)$ | Switching rate from NNRTI to DTG (maintenance therapy) | $\gamma_{S_1 \to S_3}^k(t) := 1 \text{ year}^{-1}, \ t \geq 2020$ |
| $\gamma_{F_1 \to T_3}(t), (t \geq 2020)$ | Switching rate from NNRTI to DTG (switch therapy) | $\gamma_{F_1 \to T_3}(t) = 1 \text{ year}^{-1}, \ t \geq 2020$ |

DTG, dolutegravir; MARISA, Modeling Antiretroviral drug Resistance In South Africa; NAMSAL, New Antiretroviral and Monitoring Strategies in HIV-infected Adults in Low-income countries; NNRTI, non-nucleoside reverse transcriptase inhibitor; PI, protease inhibitor.

efficacy was similar to the one observed in the New Antiretroviral and Monitoring Strategies in HIV-infected Adults in Low-income countries (NAMSAL) study [12], corresponding to an OR of failure between NNRTI and DTG of 1.02, after adjusting for the different baseline characteristics of the 2 groups (see S1 Text, Section 2.5). Other DTG efficacy corresponding to an OR of 2 and 5 were investigated in an additional analysis (see supporting information S1 Text, Section 5.3). All code and manuscript are available from https://github.com/anthonyhauser/MARISA2. This study was approved by the Ethics Committee of the Canton of Bern, Switzerland.

## Scenarios

The model investigated the impact of the introduction of DTG-based regimens on the level of NNRTI resistance in diagnosed individuals (i.e., NNRTI PDR) under 2 main scenarios, with 4 variations for each of the 2 scenarios. We also examined the scenario where DTG-based ART is not introduced. There were thus 9 scenarios in total. The 2 main scenarios were:

1. DTG is only used in first-line regimen of ART-initiators and, as second-line, in patients failing NNRTI-based ART;

2. DTG is used as initial first-line regimens (for ART-initiators), with all patients on NNRTI-based regimens being switched to a DTG-based regimen.

For the second scenario, we varied the impacts of NRTI resistance on DTG-based regimen, by either assuming an OR of DTG-failure between NRTI-resistant and NRTI-susceptible individuals of 1 (i.e., no impact) or of 2. Each scenario also investigated 4 different DTG eligibility levels $p_1$ for women. The population eligible for DTG in each scenario was:

a. only men (100% men, 0% women)

b. men and women beyond reproductive age (100% of men, 17.5% of women)

c. men and women beyond reproductive age or using contraception (100% of men, 63% of women)

d. all men and women (100% of men, 100% of women)

The percentages of women eligible for DTG in (b) and (c) were determined by analyzing cohort data from IeDEA, which show that 17.5% adult women on ART are 50 or older [19], and estimates on the use of contraception from the World Bank [22] (see S1 Text, Section 3.1). Throughout the rest of the paper, scenarios (b) and (c) will be referred as "men and women beyond childbearing age" and "men and women not at risk of pregnancy," respectively. Of note, to model scenario 1., we set $\gamma_{S_1 \to S_3}(t) = 0$ and $\gamma_{F_1 \to T_3}(t) = \gamma_{F_1 \to T_2}^{inel}(t)$ (as opposed to scenario 2., where $\gamma_{S_1 \to S_3}(t) = \gamma_{F_1 \to T_3}(t) = 1 \text{ year}^{-1}$). We thus assumed that in all scenarios DTG will be used in second-line regimens for people failing NNRTI-based first-line ART.

## Additional analyses

We predicted the impact of different levels of DTG introduction on the level of NNRTI failure. We considered the scenario where DTG was prescribed to ART initiators and those on NNRTI-based first-line ART were switched to a DTG-based regimen, with the 4 different levels of women accessing DTG-based ART (see "Scenarios"). However, we assumed that 99% of women were eligible for DTG in scenario (d) (instead of 100%), in order to estimate NNRTI failure when only a very small fraction of women rely on it. For each of these scenarios, we predicted the percentage of individuals failing an NNRTI-based regimen in 2035 after different

durations on ART (1 or 2 years). For each of the scenarios, we ran the model from 2005 up to 2035 and retained the numbers of people starting NNRTI-based first-line ART (by CD4 groups, NNRTI resistance and sex) in 2035. We then ran the model for the compartments related to NNRTI-treatment, using the previously saved starting values. This way, we could predict the levels of NNRTI-failure in patients starting NNRTI in 2035 after 1 or 2 years of ART.

We assessed the impact of different switching rates from NNRTI- to DTG-based regimens, fixed to 1 year$^{-1}$ for both suppressed and failing individuals. We varied both rates $\gamma_{S_1 \rightarrow S_3}(t)$ and $\gamma_{F_1 \rightarrow T_3}(t)$ within a range corresponding to a time to switch of between 0.5 and 10 years after start of ART. For each analysis, the percentage of women who are DTG-eligible varied from 0% to 100%.

## Sensitivity analyses

The values of 8 parameters were varied in the sensitivity analysis: 3 transmission-related parameters (percentage of men who have sex with men (MSM), probability of male-to-male infection per sexual contact, and HIV prevalence ratio between MSM and heterosexuals), 4 resistance-related parameters (resistance rates, reversion to wild-type rate, and the effect of NNRTI resistance on NNRTI efficacy), and 1 parameter related to treatment (efficacy of DTG-based treatment). Multivariate uncertainty within specified ranges was assessed using Latin hypercube sampling [23]. Each model estimate is reported with a 95% sensitivity range. Further details are available in S1 Text, Section 3.2. and Fig C and D. In addition, we also investigated the impact on NNRTI PDR of (1) lower treatment-initiation rates than suggested by the Treat-All policy (as suggested by [24]); (2) treatment interruption; and (3) higher efficacy of DTG and different impacts of NRTI resistance on DTG (S1 Text, Section 5).

## Results

### Use of NNRTIs and levels of resistance

The percentages of patients treated with DTG and NNRTI for 9 scenarios are shown in Fig 2. The predicted evolution of levels of NNRTI PDR up to 2040 across 13 scenarios is shown in Fig 3. The model predicts that while NNRTI PDR would increase substantially under continued NNRTI-based ART, the introduction of DTG-based ART can halt this increase, if in addition to starting new patients on a DTG-based regimen the patients on NNRTI-based regimens are switched to DTG-based first-line ART. Specifically, under the scenario of continued NNRTI-based ART as standard first-line therapy, NNRTI PDR would increase to 29.8% (95% sensitivity range: 7.4% to 39.4%) by 2030 and 52.4% (21.1% to 63.4%) by 2040 (Fig 3A). At the other end of the spectrum, initiating all new ART patients on DTG-based ART and rapidly switching all patients currently on NNRTI-based ART to DTG-based regimens, independently of their sex, would stabilize NNRTI PDR at a moderate level, with a prevalence of 8.4% (2% to 11.8%) by 2030 and 10.4% (4.4% to 13.8%) by 2040 (Fig 3B). When assuming an impact of NRTI resistance on DTG-failure (Fig 3C), we found slightly higher levels of NNRTI PDR: 9.7% (2.4% to 13%) in 2030 and 13.2% (5.6% to 16.9%) in 2040, but a similar impact of the different scenarios of DTG introduction. Using DTG only in first-line regimens of patients initiating ART is not sufficient to curb the increase of NNRTI PDR, even when given to all men and women (Fig 3A).

If restricted to men, DTG-based ART will not curb the increase in NNRTI PDR: The prevalence of resistance is predicted to increase over the entire study period, reaching values of close to 40% by 2040 (Fig 3B). The situation is similar under the scenario of initiating or switching

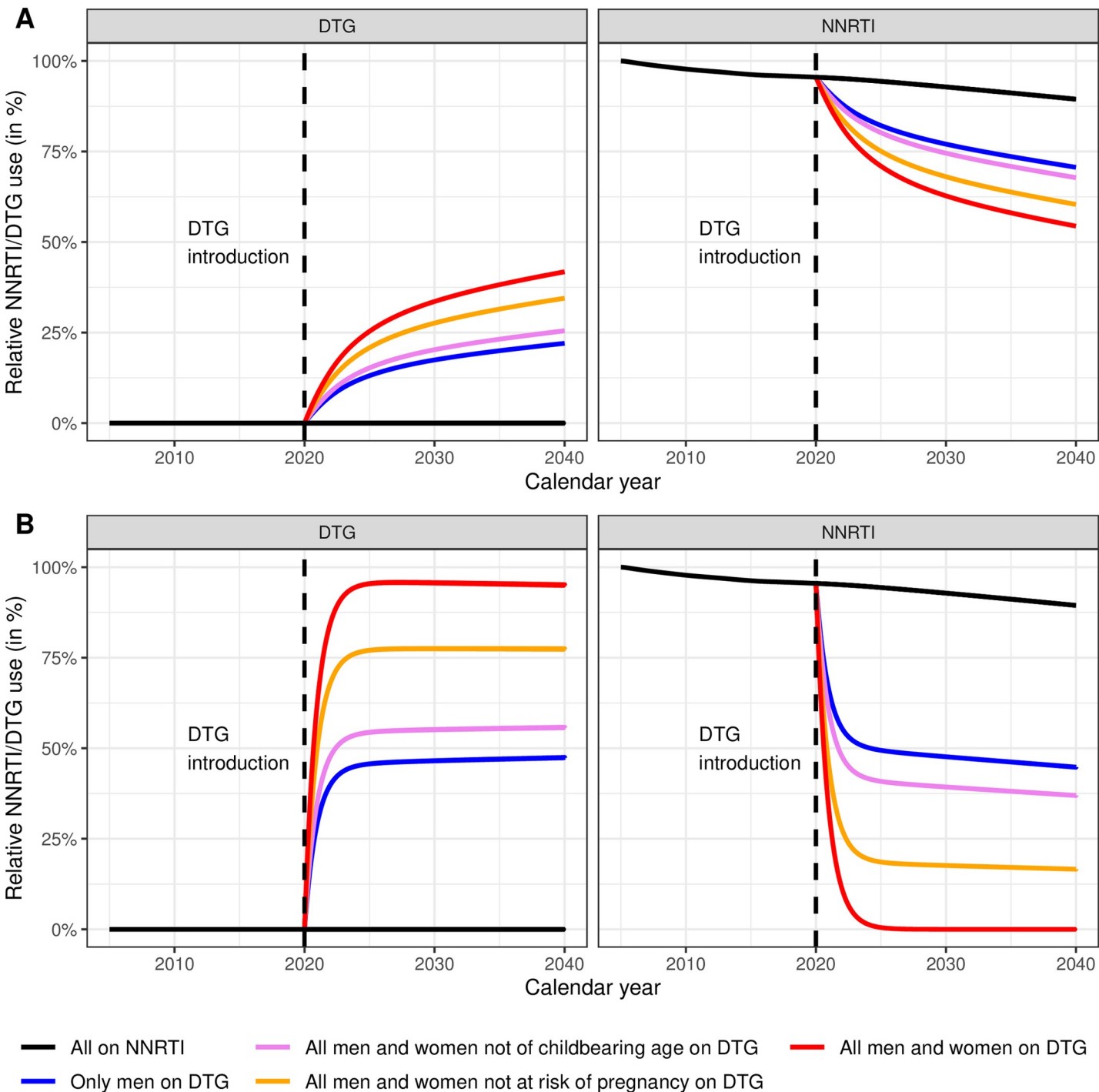

**Fig 2. Predicted use of NNRTI- and DTG-based regimens.** Percentages of patients treated with NNRTI- and DTG-based regimens (left and right panels, respectively) are shown. Panels A represent the scenarios where DTG is used in patients initiating ART, while in panels B, patients are also switched to DTG-based first line ART. ART, antiretroviral therapy; DTG, dolutegravir; NNRTI, non-nucleoside reverse transcriptase inhibitor.

men and women beyond childbearing age (17.5% of women in the IeDEA cohorts). However, the model estimates that the increase in the prevalence of NNRTI PDR is substantially slowed down if women beyond the age of reproduction or on contraception (63% of women) also

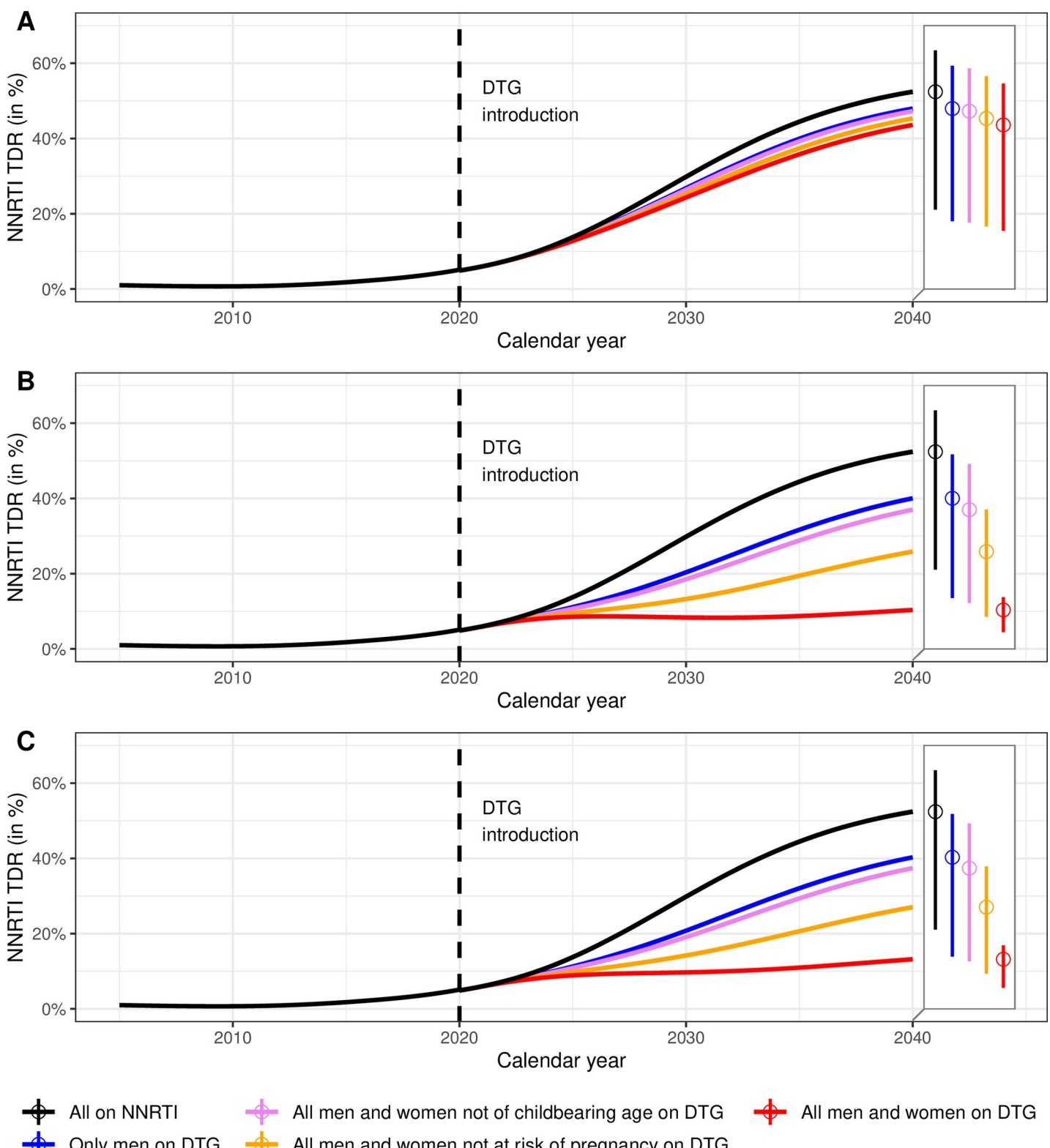

**Fig 3. Predicted levels of NNRTI PDR in South Africa 2005–2040.** DTG is introduced in 2020 under 3 scenarios: DTG as first-line regimen for ART-initiators (panel A), DTG for all patients (panel B) or DTG for all patients, assuming an impact of NRTI resistance on DTG-efficacy (panel C), and with different eligibility criteria for women (colors). The baseline model shows the situation without the introduction of DTG (black line). The 2 boxes on the right of each panel represent the levels of NNRTI PDR in 2040 and their 95% sensitivity ranges. ART, antiretroviral therapy; DTG, dolutegravir; NNRTI, non-nucleoside reverse transcriptase inhibitor; NRTI, nucleoside reverse-transcriptase inhibitor; PDR, pretreatment drug resistance.

initiate a DTG-based regimen or switch to DTG. Under this scenario, the prevalence of NNRTI PDR is predicted to reach 25.9% (8.5% to 37%) in 2040 and 13.3% (3% to 18.7%) in 2030, if DTG is given to both ART-initiators and individuals already on NNRTI-based ART (Fig 3). Again, slightly higher NNRTI PDR levels are observed when including the impact of NRTI resistance on DTG-efficacy: 27% (9.3% to 37.9%) in 2040 and 14.2% (3.4% to 19.3%) in 2030.

## Impact of switching rates

We calculated levels of NNRTI PDR for 2035 for different average switching delays and percentages of women eligible for DTG-based ART, however, without considering the effect of NRTI resistance. We considered the effect of a modified switching rate both in suppressed individuals (Fig 4A) and in individuals on a failing regimen (Fig 4B). The predicted levels of NNRTI PDR range from 8.4% to 33.1%. The results indicate potential benefits of both strategies to reduce NNRTI resistance. However, as shown by the greater variation in the prevalence of NNRTI PDR in the vertical than horizontal direction in Fig 4, allowing a higher proportion of women access to DTG-based ART has a greater impact than increasing switching rates.

## Impact of DTG-eligibility on the rate of NNRTI failure

As expected from their effect on NNRTI resistance, the different scenarios of the rollout of DTG-based ART also influence the virological failure under NNRTI-based ART. Fig 5 shows the predicted proportion of NNRTI-failure after 1 and 2 years among DTG-ineligible women according to different scenarios of DTG-introduction. In the absence of DTG introduction, we observe a high level of failure in women starting NNRTI in 2035, reaching 18.2% after 2 years of ART. If all men are started on or switched to DTG, it would help diminish the level of failure by 2 years to 14.3% in 2035, and to 14.5% when including the impact of NRTI resistance. This percentage decreases to 13.1%, when including all women not at risk of pregnancy (13.4% when including NRTI resistance), and to 12.3% (12.7% when including NRTI-resistance) if all men and 99% of women are included. Finally, we find that the increase of virological failure in DTG-ineligible women who still rely on NNRTI can be stopped by the introduction of DTG (see S1 Text Fig E).

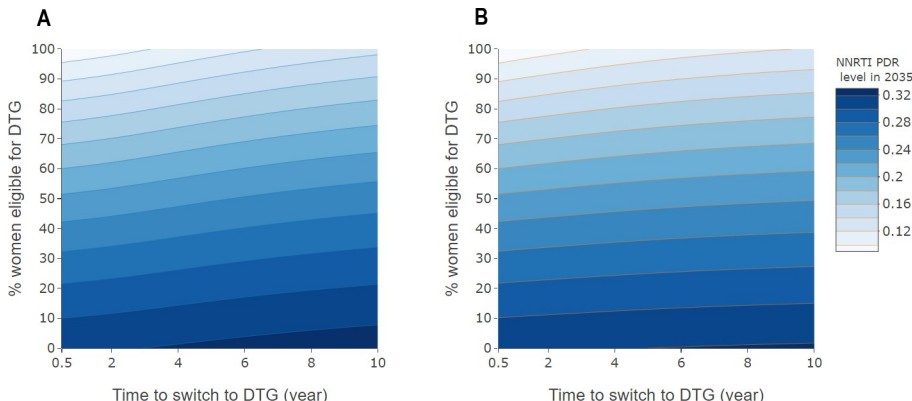

**Fig 4. Level of NNRTI PDR in 2035, by rate of switching to DTG-based ART and percent women eligible for DTG-based ART.** Panel A relates to patients on first-line ART with suppressed HIV-1 replication, and panel B to individuals failing NNRTI-based ART. The average time to switching (i.e., the inverse of the switching rate) varies from 0.5 to 10 years for individuals with viral suppression (panel A) or failure (panel B). ART, antiretroviral therapy; DTG, dolutegravir; NNRTI, non-nucleoside reverse transcriptase inhibitor; PDR, pretreatment drug resistance.

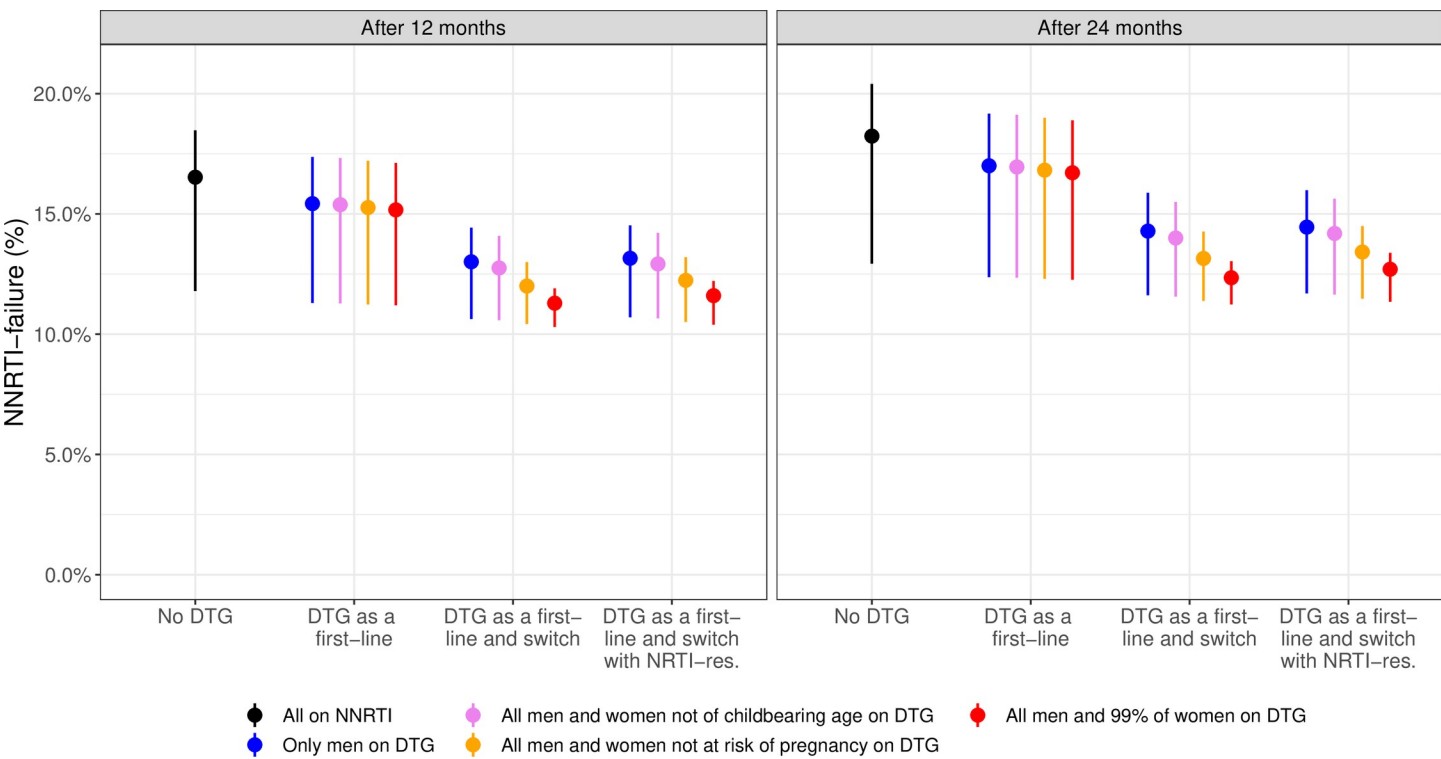

**Fig 5. Predicted percentage of women failing NNRTI-based ART after 1 and 2 years of ART in 2035, depending on the scenario of the rollout of DTG-based ART.**
Note that the scenario in which DTG is given to all men and 99% of women (in red) replaces the scenario in which DTG was given to all men and women (see "Additional analyses"). Failure is given after 1 and 2 years of ART. ART, antiretroviral therapy; DTG, dolutegravir; NNRTI, non-nucleoside reverse transcriptase inhibitor.

## Discussion

We adapted the epidemiological MARISA model to examine the impact of the scale-up of DTG-based ART on NNRTI PDR in South Africa. Overall, our findings suggest that if a large fraction of women is excluded from receiving DTG-based ART, they will not only receive a potentially inferior NNRTI-based regimen but will also face increasing rates of resistance to this regimen due to the population level effects of continued NNRTI use. In contrast, the spread of NNRTI resistance can be slowed down if DTG-based ART is made accessible both to women at low risk of pregnancy and to people currently on an NNRTI-based first-line regimen, thereby indirectly protecting those still requiring an NNRTI-based treatment. Model simulations emphasize the importance of starting on or switching a maximum number of women to DTG-based ART: Increasing use of DTG-based regimens was the strategy with the greatest potential to curb the spread of NNRTI resistance. The latter strategy will also lower the risk of virologic failure in women who have to rely on NNRTI-based ART in the future. Finally, it is interesting to observe that, even when using DTG for all patients, NNRTI PDR is not expected to decrease but rather to remain approximatively stable at a moderate level, due to the very slow reversion of NNRTI resistance that allows subsequent transmission of NNRTI resistance (in line with [25]).

While some countries, such as South Africa, first considered limiting access to DTG to men, menopausal women, and women using long-term family planning as a potential policy, the new WHO guidelines state that women should not in principle be excluded from DTG-based ART, even women who are at risk of pregnancy or desire to get pregnant. WHO recommends a woman-centered approach where women should be provided with information about

benefits and risks to make an informed choice [16,26]. It is unclear what proportion of women will effectively receive DTG-based ART, as it depends on individual women's decisions. In this context, model simulations are essential in order to assess the impact of the different options proposed and different levels of DTG uptake. A strength of our model is that it deals with the 2 most significant sources of uncertainty associated with the introduction of DTG, namely DTG uptake in women and the delay in switching people currently on NNRTI regimens. Despite the uncertainty concerning the uptake of DTG in women, it is likely that a proportion of women will continue to rely on NNRTI-based ART. Therefore, even with the rollout of DTG, NNRTI resistance will continue to be relevant for these women. Compared with other modeling work that assessed risks and benefits of DTG introduction (e.g., [15]), our model focused on its indirect, population-level impact on NNRTI resistance. Rather than assigning a level of NNRTI resistance that is fixed over time, HIV care and disease stages (as in [15]), our model considered the dynamic development of NNRTI resistance under relevant scenarios.

Our model also has several limitations. First, real-world data on the efficacy of DTG, especially in resource-limited settings, are scarce. Therefore, we conservatively assumed that DTG has a similar efficacy as observed in the NAMSAL study [12]. Higher DTG efficacies as well as different impacts of NRTI resistance on DTG-failure are investigated in supplementary analyses (see S1 Text, Section 5.3). Second, predictions of levels of NNRTI resistance over the next 20 years are naturally uncertain, as reflected by the wide sensitivity ranges in Fig 3. However, despite the uncertainty, it is clear that the different strategies of rolling out DTG-based ART influenced the levels of NNRTI resistance. Finally, the MARISA model includes some simplifying assumptions, e.g., we did not model prevention of mother-to-child transmission (PMTCT), or treatment interruption. However, relaxing some of these assumptions did not drastically change our conclusion (see S1 Text, Section 5).

Another limitation of this study is the fact that the MARISA model does not take into account resistance to DTG and uses a simplified representation of NRTI resistance. In the context of the introduction of DTG-based ART, modeling of NRTI resistance is particularly relevant as individuals starting on DTG as a functional monotherapy due to resistance to both NRTI backbones—TDF and 3TC—experience higher risk of treatment failure [27]. As they are considerably less frequently transmitted [28] and revert back quickly [29,30], NRTI resistances might primarily be an issue for ART-experienced individuals and more specifically, in patients failing NNRTI-based regimens, who often exhibit high levels of NRTI resistance [31]. These patients who are on non-suppressive NNRTI-based regimens are expected to switch to DTG, either after identification of treatment failure, following the new WHO guidelines, or blindly [17]. In the context of modeling the DTG rollout, this consideration has 2 important implications. First, patients currently failing NNRTI-based regimens are expected to have higher DTG failure rates, mainly due to previously acquired NRTI resistance. Second, due to ongoing viral replication and due to preexisting NRTI resistance, they are at higher risk of accumulating resistance, which may also lead to the emergence of DTG resistance. So far, data on emergence of DTG resistance are primarily available from patients in whom treatment failure was detected relatively early, which may not be the case in African settings [9]. Therefore, to understand risk inherent in the emergence of DTG resistance, adapting the MARISA model by extending its resistance dimension to DTG resistance will be necessary.

## Conclusions

In conclusion, our study indicates that giving access to DTG-based ART to all women not at risk of pregnancy could limit the increase of NNRTI PDR, but even if all women receive DTG-based ART, the level of NNRTI PDR will remain above 10% in South Africa. Our model

highlights the importance of a rapid switch of patients currently on NNRTI-based to DTG-based ART in order to limit the increase in NNRTI PDR. Women who remain on NNRTI-based ART will indirectly benefit from a high level of DTG uptake due to a reduced risk of virologic failure.

## Supporting information

**S1 Text. Supporting information.** Section 1: description of the adapted MARISA model. Section 2: description of model parameters. Section 3: model simulation. Section 4: model equations. Section 5: additional results and sensitivity analyses.
(PDF)

## Acknowledgments

Computations were conducted on UBELIX (http://www.id.unibe.ch/hpc), the high performance computing cluster at the University of Bern, Switzerland.

## Author Contributions

**Conceptualization:** Anthony Hauser, Matthias Egger, Roger D. Kouyos.

**Data curation:** Anthony Hauser.

**Formal analysis:** Anthony Hauser, Julien Riou, Roger D. Kouyos.

**Funding acquisition:** Matthias Egger, Roger D. Kouyos.

**Methodology:** Anthony Hauser, Katharina Kusejko, Leigh F. Johnson, Julien Riou, Roger D. Kouyos.

**Project administration:** Matthias Egger.

**Resources:** Matthias Egger.

**Software:** Anthony Hauser, Julien Riou.

**Supervision:** Matthias Egger, Roger D. Kouyos.

**Validation:** Leigh F. Johnson, Huldrych F. Günthard, Gilles Wandeler.

**Writing – original draft:** Anthony Hauser, Matthias Egger, Roger D. Kouyos.

**Writing – review & editing:** Katharina Kusejko, Leigh F. Johnson, Huldrych F. Günthard, Julien Riou, Gilles Wandeler, Matthias Egger, Roger D. Kouyos.

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
