## [Decision Letter · Decision Letter 0]

9 May 2020

Dear Matthias,

Thank you very much for submitting your manuscript "Impact of Scaling up Dolutegravir on Antiretroviral Resistance in South Africa: A Mathematical Modelling Study" (PMEDICINE-D-19-03520) for consideration at PLOS Medicine. 

Your paper was evaluated by a senior editor and discussed among all the editors here. It was also discussed with an academic editor with relevant expertise, and sent to independent reviewers, including a statistical reviewer. 

My sincere apologies that this paper took longer to peer-review than would be usual (or hoped-for); unfortunately despite the policy value of the analyses presented, it was difficult to sign up qualified reviewers with good modelling expertise in this case. However, we do now have three valuable reviews all of whom recognise the potential of the paper for PLOS Medicine and, we hope, make useful comments that will help to refine the article for potential publication. Please see the reviews below. 

In addition to the reviews, the Academic Editor for your article commented that it may be valuable to update the text and place it in the current HIV programmatic context. Given the IeDEA team has cohorts across all over the world, countries will be able to communicate DTG eligibility for women of reproductive potential. The AE noted that in South Africa, updated guidelines were set out in Oct 2019 by the South Africa Department of Health, link: https://sahivsoc.org/Files/2019%20Abridged%20ART%20Guidelines%2010%20October%202019.pdf ). These guidelines suggest that South Africa has proposed DTG in adult and adolescent males and opted for a women-centred approach where women are informed of benefits and risks and decide on their regimen. For females < 6 weeks into their pregnancy and females planning to get pregnant in the future, efavirenz appears to be preferred. For females >6 weeks into their pregnancy and females not pregnant and not planning to get pregnant, DTG is preferred. Some of the discussion in the article could potentially be updated in this respect. 

The reviews are appended at the bottom of this email and any accompanying reviewer attachments can be seen via the link below. Please also see very minor formatting/writeup changes from the editors (below, above the reviewers' comments). 

[LINK]

In light of these reviews, I am afraid that we will not be able to accept the manuscript for publication in the journal in its current form, but we would like to consider a revised version that addresses the reviewers' and editors' comments. Obviously we cannot make any decision about publication until we have seen the revised manuscript and your response, and we plan to seek re-review by one or more of the reviewers. 

We expect to receive your revised manuscript by Jun 01 2020 11:59PM. Please email us (plosmedicine@plos.org) if you have any questions or concerns.

We look forward to receiving your revised manuscript. 

Sincerely,

Emma Veitch, PhD

PLOS Medicine

On behalf of Clare Stone, PhD, Acting Chief Editor,

PLOS Medicine

plosmedicine.org

*Trivial request, but journal structure for abstract subsections should be Background, Methods and Findings, Conclusions (Methods and Findings a single subsection).

* In the last sentence of the Abstract Methods and Findings section, we'd suggest including a brief description of any key limitation(s) of the study methods/findings.

*At this stage, we ask that you include a short, non-technical Author Summary of your research to make findings accessible to a wide audience that includes both scientists and non-scientists. The Author Summary should immediately follow the Abstract in your revised manuscript. This text is subject to editorial change and should be distinct from the scientific abstract. Please see our author guidelines for more information: https://journals.plos.org/plosmedicine/s/revising-your-manuscript#loc-author-summary

Comments from the reviewers:

Reviewer #1: A modelling study to predict the impact of various scenarios of DTG-based ART rollout in South Africa on the levels of pretreatment NNRTI-resistance. The model uses the established MARISA model based on 5 HIV cohorts. The work is well performed with many relevant scenarios being modelled, suggesting that DTG rollout for all will lead to a substantial drop in NNRTI-resistance.

Major comment: L239-259: The authors here describe as limitations what I find a major omission of the model, or a missed opportunity. The impact of co-existing NRTI-resistance on DTG resistance development, (long-term) DTG effectiveness in real-life programmatic conditions, the influence of non-B subtypes on DTG resistance patterns, the lack of stringent VL monitoring allowing for ongoing VF, the resultant potential of transmission of DTG-resistant trains, who in turn can compromise first-line DTG-based regimens. In my view, the current model has taken a fairly narrow approach focusing on NNRTI resistance as the main outcome; it would be more useful and timely to undertake a serious attempt (even given limited available data) to include the broader outcomes connected to DTG effectiveness and resistance. 

Minor comments:

Adding a scenario of transitioning ALL patients (1st and 2nd line) to DTG would be useful.

Line 150: "Specifically, under the scenario of continued NNRTI-based ART as standard first-line therapy, NNRTI resistance would increase to 46.8% (95% sensitivity range: 19.7%-54.5%) by 2030 and 58.5% (32.5%-68.9%) by 2040 (Fig 3A). In my view, it would be unrealistic to assume that these high levels of NNRTI resistance would be accepted without any policy actions, either in terms of genotypic resistance testing, enhanced VL monitoring, or accelerated access to DTG. So the scenario modelled is too simplistic.

L156: "would stabilize NNRTI resistance at a low level, with a prevalence of 14.3% (3.5%-17.5%) by 2030 and 14.8% (6.6%-19.5%) by 2040 (Fig 3B)."

Actually, these levels of NNRTI resistance are not so low at all, and would still warrant a guide line change to standard non-NNRTI first-line according to current WHO guidelines. 

Line 21: " at risk of pregnancy" epidemiologist jargon, better say "of childbearing potential"

Reviewer #2: Impact of Scaling up Dolutegravir on Antiretroviral Resistance in South Africa: A Mathematical Modelling Study

The manuscript by Hauser et al., describes how the scale-up of dolutegravir (DTG) will impact HIV drug resistance, with specific interest on non-nucleoside reverse transcriptase inhibitors (NNRTIs). Although modelling studies have their limitations, they are useful in predicting possible outcomes based on current knowledge. Hauser et al., performed a comprehensive analysis based on the MARISA model and highlighted some very important results, especially with regards to use of DTG in women of child-bearing potential. 

General Comments: 

1. This paper clearly highlights that increasing access to DTG for women has a great impact on curbing NNRTI resistance, even when compared to increasing the rate of switching ART. 

2. However, and as mentioned at the end of the discussion section, it would be useful in future analysis to extend the models resistance dimension to NRTI resistance, as NRTIs are the main drug class administered together with DTG. Compromised NRTI drugs (such as tenofovir) will pose a risk of DTG functional monotherapy.

3. As the manuscript describes pretreatment NNRTI resistance, and in some cases NNRTI resistance at virological failure, I would suggest the authors refrain from using the term NNRTI resistance when referring to the pretreatment time-point. Rather the authors should consider using the following terms consistently:

 a. NNRTI PDR; when referring to NNRTI pretreatment resistance, and 

 b. NNRTI resistance; when referring to resistance at virological failure.

4. Overall, the manuscript is well written and was a pleasure to read.

Specific Comments

Abstract

Background

1. "We used mathematical modelling to examine the impact of the scale-up of DTG-based ART on NNRTI pre-treatment drug resistance (PDR) in South Africa, 2019-2040." 

I would suggest the authors use the term "predict" rather than "examine". 

Methods and results

2. "If all men and women beyond reproductive age or on contraception are started on or switched to DTG-based ART, NNRTI resistance would reach 35.1% in 2040." 

I think here the authors are trying to say, "If ONLY men and women beyond reproductive age or on contraception are started on or switched to DTG-based ART, NNRTI resistance would reach 35.1% in 2040." If so, please consider including the term only so that it distinguishes this group of people from the ones in the preceding sentence. 

Conclusions

3. "Starting or switching all men and women to DTG would lead to a sustained decline in resistance levels whereas using DTG-based ART in all men, or in men and women beyond childbearing age, would slow down the increase in levels of NNRTI resistance."

Consider saying, "… would only slow down the increase in levels of NNRTI resistance."

Introduction

1. "Again, the rate of switching to DTG-based second-line ART will vary, influenced by concerns about the development of dolutegravir resistance in patients who switch with pre-existing resistance to NRTIs [17]." (Line 34).

The abbreviation for dolutegravir should be used as it has already been introduced previously. 

2. "We adapted the MARISA model (Modelling Antiretroviral drug Resistance In South Africa) to examine the impact of different scenarios regarding the scale-up of DTG-based ART on NNRTI pre-treatment drug resistance ("NNRTI resistance" in the remaining of this article) in South Africa for 2019-2040." (Line 41).

As mentioned previously, I would suggest the authors use the term "predict" rather than "examine". 

Materials and methods

3. "We added and modified parameters in order to model the introduction of DTG. We assumed that the DTG initiation rate γD→T3(t) is the same as the NNRTI initiation rate γD→T1(t) from 2019." (Line 76).

Here the assumption is that the rate at which DTG will be introduced is the same as that of NNRTIs. This assumption on the initiation rate could be limited in that less people are expected to be viraemic and to transmit HIV following the introduction of DTG, due to higher viral suppression rates with DTG compared to NNRTIs. If interpreted correctly, I would suggest the authors consider including this assumption as a limitation of the study. 

Results

4. "At the other end of the spectrum, initiating all new ART patients on DTG-based ART and rapidly switching all patients currently on NNRTI-based ART to DTG-based regimens, independently of their gender, would stabilize NNRTI resistance at a low level, with a prevalence of 14.3% (3.5%-17.5%) by 2030 and 14.8% (6.6%-19.5%) by 2040" (Line 152).

14.8% (6.6%-19.5%) NNRTI PDR with no use of NNRTIs at all still seems a bit high. Please check to make sure the model approximations are correct. 

Again as mentioned in the general comments, the authors should be clear that they are referring to NNRTI PDR when they say NNRTI resistance. 

5. "Restricting DTG-based ART to men to avoid the risk of DTG-associated neural tube defects in newborns will not curb the increase in NNRTI resistance: the prevalence of resistance is predicted to increase over the entire study period, reaching values of close to 50% by 2040."

Neural tube defects should be abbreviated as NTDs as introduced before in the introduction (Line 17). NNRTI resistance should be changed to NNRTI PDR as previously suggested, and in any instances where the authors refer to pretreatment NNRTI resistance. 

6. "As expected from their effect on NNRTI resistance, the different scenarios of the rollout of DTG-based ART also influence the rate of virological failure in women NNRTI-based ART among DTG-ineligible." (Line 181). 

I think the word "on" is missing here, i.e. "…virological failure in women on NNRTI-based ART…" Besides that, this sentence is a bit confusing and the authors should consider rephrasing. 

Discussion

7. "Overall, our findings suggest that if a large fraction of women is excluded from receiving DTG-based ART, they will not only receive a potentially inferior NNRTI-based regimen but will also face increasing rates of resistance to this regimen due to the population level effects of continued NNRTI use." (Line 195).

This is a very important statement, but I agree with it partially. This is because levels of NNRTI PDR in women will inevitably decrease if all men are switched to DTG. So chances are that we will no longer worry about NNRTI PDR among women, especially in South Africa where it has been shown that most NNRTI PDR is due to heterosexual transmission. So the non-inferiority of the NNRTIs will play a major role in those that already got transmitted NNRTI mutations, but the effect will likely reduce drastically in later years with the reduction of NNRTI resistance among men due to DTG use. Just giving a thought.

8. "Model simulations emphasize the importance starting on or switching a maximum number of women to…"

I think the sentence is missing the word "of", i.e. "Model simulations emphasize the importance of starting on or switching a maximum number of women to…"

Reviewer #3: 

This is a modelling study concerned with an interesting issue, upscaling the use of Dolutegravir. While the paper is well-written and carefully conducted, some methodological issues related to the treatment of uncertainty are unclear and require clarification. Specific comments are given below.

This article is essentially a large extrapolation exercise into the future. Extrapolations are potentially dangerous and should be made with caution, placing particular attention to uncertainty representation. Here the authors conduct a multivariate (multi-way) sensitivity analysis and this is appropriate. In addition, they depict uncertainty in their figures in a clever way, thus avoiding cluttering in the main part of the figure. 

However, the analysis is based on deterministic models which, depending on their treatment, tend to underestimate uncertainty. To this end it would be useful to add the evolution of uncertainty over time. Also, it would be useful to report the uncertainty of each parameter used. For example, on table 1 and table 1 of the supplement they could add the parameter value, it's range and any distributional assumptions made for each parameter. For the calibration parameters (supplement table 1) this is vital since the results of any survival analysis should incorporate the associated uncertainty.

This issue also relates to the selected range of the sensitivity analysis, how were the numbers on supplement table 4 informed, were they evidence-based or assumption-based?

Finally, it would be good to partly amend the discussion of the results, fully reflecting the uncertainty of the findings, including the absence of "statistically significantly different" results, which simply give a scientifically honest picture reflecting the inherent uncertainty of such modelling exercises.

The design of this modelling study crucially depends upon the selected scenarios. One plausible option is to look extensively into the consequences of using NNRTI-based ART as first line treatment and DTG-based ART as second line. Presumably the evidence and long-term knowledge of the genetic interactions this may cause and the possibility of future DTG resistance development is relatively scarce? It appears that these issues deserve further discussion.

[LINK]

---

## [Decision Letter · Decision Letter 1]

26 Aug 2020

Dear Dr. Egger,

Thank you very much for re-submitting your manuscript "Impact of Scaling up Dolutegravir on Antiretroviral Resistance in South Africa" (PMEDICINE-D-19-03520R1) for consideration at PLOS Medicine.

I have discussed the paper with editorial colleagues and our academic editor, and it was also seen again by one reviewer. I am pleased to tell you that, provided the remaining editorial and production issues are dealt with, we expect to be able to accept the paper for publication in the journal.

[LINK]

Please let me know if you have any questions. Otherwise, we look forward to receiving the revised manuscript shortly. 

Sincerely,

Richard Turner, PhD

rturner@plos.org

Requests from Editors:

Please revisit your data statement and state how parameter details will be made available by the time of publication (preferably in the form of supplementary files). We suggest modifying the statement to note that you are not able to distribute prior clinical data used in the study (assuming this is correct) and that inquiries should be made to the IeDEA group about this, for example.

To your title, please add a study descriptor following a colon, e.g., "...: a modeling study". 

In your abstract, we suspect the text should be amended to "... a DTG-based regimen ...". Just before this, we suggest rephrasing " ... both ART initiators" (we think that the word "both" can be omitted). 

We suggest "substantial uncertainty" rather than "high uncertainty". 

In the sixth summary point, please make that "predicted effect of dolutegravir". 

Please substitute "sex" for "gender" where appropriate, e.g., at line 143. 

At line 237, please substitute "all patients", or similar, for "everybody". 

At line 240, please make that "... considered limiting access ...". 

Please spell out the author group name for reference 12; and format the author list for reference 25 as for other references. 

Can the title for reference 26 be converted to lowercase text?

Comments from Reviewers:

*** Reviewer #2: 

The authors addressed all comments raised. I have no further comments.

***

[LINK]

---

## [Editor Report · Decision Letter 2]

10 Nov 2020

Dear Prof. Egger, 

On behalf of my colleagues and the academic editor, Dr. Amitabh Bipin Suthar, I am delighted to inform you that your manuscript entitled "Impact of Scaling up Dolutegravir on Antiretroviral Resistance in South Africa: a modeling study" (PMEDICINE-D-19-03520R2) has been accepted for publication in PLOS Medicine. 

PRODUCTION PROCESS

Before publication you will see the copyedited word document (within 5 business days) and a PDF proof shortly after that. The copyeditor will be in touch shortly before sending you the copyedited Word document. We will make some revisions at copyediting stage to conform to our general style, and for clarification. When you receive this version you should check and revise it very carefully, including figures, tables, references, and supporting information, because corrections at the next stage (proofs) will be strictly limited to (1) errors in author names or affiliations, (2) errors of scientific fact that would cause misunderstandings to readers, and (3) printer's (introduced) errors. Please return the copyedited file within 2 business days in order to ensure timely delivery of the PDF proof. 

If you are likely to be away when either this document or the proof is sent, please ensure we have contact information of a second person, as we will need you to respond quickly at each point. Given the disruptions resulting from the ongoing COVID-19 pandemic, there may be delays in the production process. We apologise in advance for any inconvenience caused and will do our best to minimize impact as far as possible.

EARLY VERSION

PRESS

PROFILE INFORMATION

Thank you again for submitting the manuscript to PLOS Medicine. We look forward to publishing it. 

Best wishes, 

Richard Turner, PhD

Senior Editor 

PLOS Medicine

plosmedicine.org